# The Influence of Silver Nanoparticle Form on the Toxicity in Freshwater Mussels

Joelle Auclair [1], Caroline Peyrot [2], Kevin J. Wilkinson [2] and François Gagné [1,*]

1 Aquatic Contaminants Research Divsion, Environment and Climate Change Canada, 105 McGiil, Montreal, QC H2Y 2E7, Canada; joelle.auclair@ec.gc.ca
2 Chemistry Department, Montreal University, Montreal, QC H2V 2B8, Canada; caroline.peyrot@umontreal.ca (C.P.); kj.wilkinson@umontreal.ca (K.J.W.)
* Correspondence: francois.gagne@canada.ca

**Abstract:** The contribution of the form of silver nanomaterials (nAg) towards toxicity in aquatic organisms is not well understood. The purpose of this study was to examine the toxicity of various structures (sphere, cube and prism) of nAg in *Dreissena bugensis* mussels. Mussels were exposed to increasing concentrations of polyvinyl-coated nAg of the same size for 96 h at 15 °C. They were then analyzed for biophysical changes in the cytoplasm (viscosity, protein aggregation and lipids), neuro-activity (fractal kinetics of acetylcholinesterase (AChE)), oxidative stress (labile zinc (Zn) and lipid peroxidation) and inflammation (arachidonate cyclooxygenase). Although some decreasing effects in protein aggregation were observed, viscosity was more strongly decreased in mussels exposed to spheric and prismatic nAg. The activity of AChE was significantly decreased in the following form-dependent manner: prismatic > cubic > spheric nAg. The fractal dimension of AChE reactions was reduced by all geometries of nAg, while dissolved Ag had no effects. For nanoparticles with the same coating and relative size, spheric nAg produced more significant changes towards the fractal dimension of AChE, while prismatic nAg increased both protein aggregation and viscosity, whereas cubic nAg decreased protein aggregation in the cytoplasm. It is concluded that the geometries of nanoparticles could influence toxicity in aquatic organisms.

**Keywords:** viscosity; protein aggregation; acetylcholinesterase; fractal dimension; quagga mussels

## 1. Introduction

The nanotechnology industry has grown exponentially over the last decades and pervaded many areas or our economy [1]. Silver nanoparticles (nAg) are frequently found in consumer products and medical equipment because of their biocidal properties [2]. The increasing use and disposal of nAg-contaminated products have led to the release of nAg into the environment, which could lead to harmful effects in wildlife [3]. Ag is mainly released by municipal wastewaters into the aquatic environment [4]. A previous study revealed that most Ag particles at the nanoscale (1–100 nm) were effectively removed (95%) but significant amounts were still released in effluents, albeit at the low ng/L range [5]. Ag nanoparticles consisted of <8% of particulate Ag in the investigated wastewaters.

Nanoparticles come in many sizes (1–>100 nm and aggregates), coatings and forms (cubes, spheres, rods), which complicates the environmental risk for these nanomaterials. The morphology of nanomaterials was shown to influence persistence in aquatic environments [6]. When cubic nAg was dissolved in freshwater and seawater, the corners of the cubes became more rounded and formed large aggregates during aging. The release of Ag ions diffused from both the edges and the faces of the cube and aggregation was enhanced in seawater. Spheric, cubic and rod-like nanoparticles could be selectively transformed to triangular prisms by reaction with citrate/$H_2O_2$ [7]. Oxidative etching by $H_2O_2$ of spheric nAg occurred more quickly than prismatic nAg, showing form-dependent reactiv-

ity, where triangular prisms are more resilient to oxidation, perhaps due to the reduction of surface area.

The shape of Ag oxide nanoparticles influenced the bactericidal toxicity to *Escherichia coli* [8]. Cubic nAgO were more toxic to bacteria than octahedral nAgO. The influence of nAg geometries (cubes, spheres, wires and triangles) towards bacterial toxicity involved the formation of protein corona formation [9]. Triangular and spheric nAg were more toxic to *E coli* bacteria than wired and cubic nAg. The protein composition of the corona differed according to the geometries of nAg, cubic nAg containing a more diverse array of proteins, such as albumin, ferritin and collagenic chains, than the other forms of nAg. Protein corona thickness was lower in the most toxic forms of nAg, indicating a tenuous protein shell on the nAg exposing Ag more easily at the surface to the surroundings. The toxicity of various shapes of polyvinylpyrrolidone-coated nAg at the same size range was recently examined in freshwater mussels [10]. The study revealed that similarly coated and similar sizes of spheric, cubic and prismatic nAg but not ionic Ag produced different effects. At the level of liquid crystal formation in cells, the following form of nAg produced changes: prismatic > cubic > spherical nAg. A reduction in the fractal diffusion rate of the pyruvate kinase–lactate dehydrogenase complex was observed, leading to decreased specific activity and increased protein damage/denaturation based on protein–ubiquitin patterns. Hence, the increase in viscosity and formation of liquid crystals could initiate protein damage and lead to toxicity. Given the increasing production of nAg forms (wires, octohedra, cubes, spheres, prisms, triangles) essentially for bacteria and fouling prevention of various devices and clothes, further research on their toxicity is warranted. The biophysical changes of protein and lipid status in cells require further examination with respect to protein aggregation and soluble lipids in the cytoplasm. Protein aggregation is involved in aging (lipofuscins) and plaque formation in some degenerative diseases [11,12]. Neurotoxicological effects of nanomaterials are of concern given the importance of membrane integrity for neurotransmission, namely, choline signaling and acetylcholinesterase (AChE) activity [13]. This enzyme is often used to measure neural activity in aquatic organisms. In fish embryos, a dose-dependent inhibition of AChE activity was observed at concentrations as low as 1 µg/L nAg. The study also revealed that inhibitions in AChE impaired cholinergic signaling and led to impaired immunity. The activity in AChE was also significantly depleted in *Ruditapes philippinarum* clams exposed to nAg [14]. Indeed, AChE was decreased five-fold in clams exposed to 10 µg/L nAg for 14 days.

This study sought to compare the toxicity of ionic Ag and nAg of varying geometries (spherical, cubic and prismatic) in mussels. The nAg had the same coating (polyvinylpyrrolidone) and the same dimensions to highlight the effects of the form of nAg. Viscosity, protein aggregation and lipid mobilisation were examined as the first line of impacts of nanoparticles, followed by AChE activity. Since nanoparticles introduce biophysical changes in cells, such as viscosiy, liquid crystals and protein aggregation, the fractal dimension (*fD*) of AChE activity was examined in the soft tissues of freshwater mussels. The increase in viscosity in the cytoplasm of cells could induce crowding effects and disrupt the normal functioning of enzymes and lipids, leading to toxicity.

## 2. Methods

### 2.1. Mussel Exposure Experiments

*Dreissenna bugensis* mussels were collected by hand at Forth Lennox in Richelieu River (Île-aux-Noix, QC, Canada) in June in 2019 and shipped in cooled containers under towels wet with river water. They were allowed to stand in a 60 L (500 mussels) aquarium under constant aeration for one month before the experiments. They were fed 2–3 times a week with commercial coral reef feed supplemented with *Pseudokirchneriella subcapitata* algae (circa 500 million algae per aquarium) during that time. Mussels were thus exposed to polyvinylpyrrolidone (PVP) (NanoXact, from nanoComposix Inc, San Diago, CA, USA) of similar size (70–80 nm) with the geometry shown in Table 1. Mussels were exposed to

2 concentrations of dissolved Ag (2 and 10 μg/L from $AgNO_3$) and 3 concentrations of nanoparticle suspensions (2, 10 and 50 μg/L as total silver). These concentrations were chosen based on a previous study showing changes in AChE activity in clams exposed to nAg [14]. The mussels were exposed for 96 h at 15 °C with one medium change after 48 h. Total Ag levels and the size distribution of nAg were checked in the aquarium water after 1 dissolution. Suspensions of nAg were prepared in MilliQ water before making the nominal exposure concentration of 10 and 50 μg/L in aquarium water. The aquarium water consisted of UV-treated and charcoal-filtered tap water. The supplier provided the physical/chemical characteristics (transmission electron microscopy images and particle size distributions) of purchased forms of nAg [15].

**Table 1.** Silver nanoparticle properties.

| Name | Coating | Size (nm) | Surface Area $(nm^2)$ | Zeta Potential (mvolt) |
| --- | --- | --- | --- | --- |
| Prismatic nAg | PVP | 78 ± 10 (equilateral prism) | 6000 | −40 |
| Spheric nAg | PVP | 72 ± 7 (diameter) | 31,416 | −45 |
| Cubic nAg | PVP | 75 ± 7 (edge) | 60,000 | −36 |

PVP: polyvinylpyrrolidone. Silver NPs were prepared at concentrations of 2, 10 and 50 ug/L in aquarium water. Na: not determined.

Following the exposure period, mussels were transferred in clean aquarium water for 3 h. A subgroup of mussels were kept aside for air survival stress assessments. The mussels were weighed, shell length determined and the tissues were dissected and placed on ice. Tissues were weighed to obtain the condition index (CI: mussel weight/shell length).

The tissues were minced in 100 mM NaCl containing 25 mM HEPES-NaOH (pH 7.4), 1 μg/mL aprotinin protease inhibitor) and 0.1 mM dithiothreitol, then grinded using a Teflon pestle tissue grinder (5 passes) at 4 °C. A fraction of the homogenate was centrifuged at 15,000× *g* for 30 min at 2 °C for the supernatant (S15 fraction). The homogenates and S15 fractions were then stored at −85 °C. The levels of proteins in the homogenate and S15 fractions were assayed using the Bradford assay with serum bovine albumin for quantitation [16].

## 2.2. Resistance to Air Emersion

The capacity of mussels to resist air emersion was assessed to determine the general health status of mussels following exposure to forms of Ag [17]. Following exposures, mussels (*N* = 10) were randomly selected and the condition factor (mussel weight/longitudinal shell length) determined. Mussels were placed individually in plastic boats and placed in an incubator at 20 °C at 80% saturation humidity. Mussels were checked daily for survival (open shells or absence of muscular tension) and were weighed to determine the dehydration rate given that the initial mass loss comes from water. Air survival data was expressed as the mean lethal time (days) and the dehydration rate was obtained by $100 \times (weight_{T0} - weigh_{Tdeath})/weight_{T0}$.

## 2.3. Acetylcholinesterase Activity

The activity in acetylcholinesterase (AChE) was determined using acetylthiocholine as the substrate analogue in the spectrophotometric assay as previously described [18]. A 25 uL sample of the 15,000× *g* supernatant was combined with 150 uL of 0.1 mM acetylthiocholine in 0.2 mM 5,5′-dithiobis-(2-nitrobenzoic acid) (DTNB) in 50 mM Tris-acetate pH 8.2. Absorbance (412 nm) was measured every 30 s for 60 min in a microplate reader under flash mode (Synergy-IV, Biotek Instruments, Winooski, VT, USA). Enzyme activity was determined under these conditions and long-term memory effects were analysed

to determine the fractal dimension (*fD*) of the reaction rates [19]. The *fD* is a measure in the dimension reduction owing to spatial changes induced by nanoparticles and it is based on the fractal organization of complex protein networks in the cytoplasm. The *fD* is calculated by the rescaled range analysis with the Hurst coefficient: $fD = 1/H$ [20]. The Hurst coefficient represents the slope between absorbance changes in time on a log–log scale: log Absorbance = $H$ log time + constant. Unrestricted free diffusion corresponds to $H = 0.5$, giving an $fD = 2$ corresponding to diffusion in Euclidian space. Decreases in the *fD* or increased H indicates reduced dimensionality of the relevant process.

### 2.4. Oxidative Stress and Inflammation

Levels of lipid peroxidation (LPO) and the release of labile Zn were also determined as markers of oxidative stress [21,22]. Lipid peroxidation in the homogenates was determined by the thiobarbituric acid reactants procedure [23]. Standards of malonaldehyde (tetramethoxypropane) were used for calibration. Data were expressed as µg TBARS/mg protein. The levels of labile zinc were determined by the fluorescence probe methodology [24]. Briefly, a sample of the S15 fraction was prepared in 150 µL of 10 µM TSQ probe in 20% dimethylsulfoxide (DMSO) in phosphate buffered saline. Fluorescence at 360 nm excitation and 490 nm emission was read in dark microplates. Standards preparation of $ZnSO_4$ was performed. The data were expressed as relative fluorescence units/mg proteins). Activity of arachidonate-dependent cyclooxygenase (COX) activity was also determined in soft tissues extracts as previously described [21]. The detection of formed $H_2O_2$ was achieved by mixing with dichlorofluorescein and horseradish peroxidase reagent. The S15 fraction was incubated with 50 µM arachidonate, 2 µM of dichlorofluorescein and 0.1 µg/mL horseradish peroxidase in 50 mM Tris-HCl, pH 8.0, and 0.05% Tween-20 for 20 min at 20 °C, and fluorescence (485 nm excitation/528 nm emission) was measured every 5 min in dark microplates (Synergy 4, Biotek microplate reader, Winooski, VT, USA). The data were expressed as the increase in fluorescence/min/mg proteins.

### 2.5. Protein Aggregation and Total Lipids

Protein aggregation is a precondition leading to protein plaque formation and was determined in tissues based on the thioflavine T fluorescence methodology [25]. The post-mitochondrial fraction (S15) was mixed in 100 µM thioflavine T for 10 min at 20 °C. Fluorescence (450 nm excitation/480 nm emission) was measured in 96-well dark microplates. Controls of β-amyloid proteins (associated with Alzheimer's disease) were used following 2 h incubation with 0.1 µg/mL of 50 nm diameter polystyrene nanoparticles. The data were expressed as relative fluorescence units (RFU)/mg proteins. Neutral lipids levels were determined using the fluorescent Nile red stain [26]. Nile red was diluted in water at 10 µM and was added to the S15 fraction for 5 min before fluorescence (500 nm excitation/660 nm emission) readings. Solutions of Triton X100 were used as standards. The data were expressed as relative fluorescence units (RFU)/mg proteins.

### 2.6. Data Analysis

Mussels were exposed to 2, 10 and 50 µg/L of nAg forms and to 2 and 10 µg/L of ionic or dissolved Ag in 3 replicates of N = 4 mussels each. The size range of the nanoparticles was 50 nm and all were coated with polyvinylpyrolidone (PVP) to minimize influences from the geometry of the nanoparticles. Non-parametric analysis of variance (based on ranks) was performed on the data and differences between treatments were determined by the Conover–Inman test. The Pearson product–moment procedure was used for correlation analysis. Significance was set at $p \leq 0.05$ using the SYSTAT software package version 13 (USA).

## 3. Results

Mussels were exposed to three concentrations (2, 10 and 50 µg/L) of different forms of nAg (spheres, cubes and prisms) and two concentrations of ionic Ag (1 and 10 µg/L). The

dimensions of the nAg were in the same range and all were coated with PVP to maximize the influence of nAg geometry (Table 1). The size ranges of the nAg forms were 70 nm diameter for the spheres, 75 nm for the edges of the cubes and 90 nm for the equilateral prisms based on TEM analysis from the supplier. The Zeta potentials of the PVP-coated nAg were in the same range ($-42$ to $-37$ mvolt) in MilliQ water. In the exposure media (aquarium water), the Zeta potential was $-7.5$ mvolts with no evidence of precipitation (i.e., when the Zeta potential nears 0 mvolt). This information led us to prepare the dilutions of the PVP-coated nAg forms at 1 mg/L in MilliQ water prior to the final dissolution in aquarium water. The nominal concentration of Ag was determined after 1 h dissolution and was between 70–90% of the theoretical Ag concentration for each form of Ag, thus confirming the stability of nAg suspensions during the initial exposure to mussels. The hydrodynamic diameter/edge size remained stable after 24 h dissolution in aquarium water. Although they were similar in size and coatings, the surface areas (SA) differed between geometries and were estimated based on a perfect sphere ($4\pi r^2$, $r$ = radius in nm), cube ($6e^2$, $e$ = edge in nm) and prism ($2 \times base\ perimeter \times h$), assuming a height of 10 nm based on TEM analysis. The SA of the nAg was in decreasing order: prims < sphere < cube at the same concentration. The calculated SA controlled for concentration revealed that the 10 µg/L spherical nAg (31,000 nm$^2$) was similar to the prismatic nAg at 50 µg/L (30,000 nm$^2$). Hence, these two groups could be used to verify the hypothesis that SA area is a contributing factor to the geometry of nAg (a significant difference would suggest that other factors than SA were at play).

Mussel health status was determined by following the condition factor (mussel weight/shell length) and air survival time following exposures to the various forms of Ag (Figure 1). The data revealed low changes in these responses in exposed mussels, with the exception of a small increase in the condition factor in mussels exposed to 50 µg/L cubic nAg and somewhat decreased air survival time in mussels exposed to 2 µg/L prismatic nAg. Air survival time dropped from 7 days in controls to 6 days in mussels exposed to the low concentration of prismatic nAg. This suggests that the exposure concentration and Ag forms did not have acute lethal impacts on the mussels. Based on added SA, no significant differences were obtained for CF and air survival between nAg of similar SA (10 µg/L spherical and 50 µg/L prismatic nAg), suggesting no other influence (form, reactivity) contributed to the observed effects.

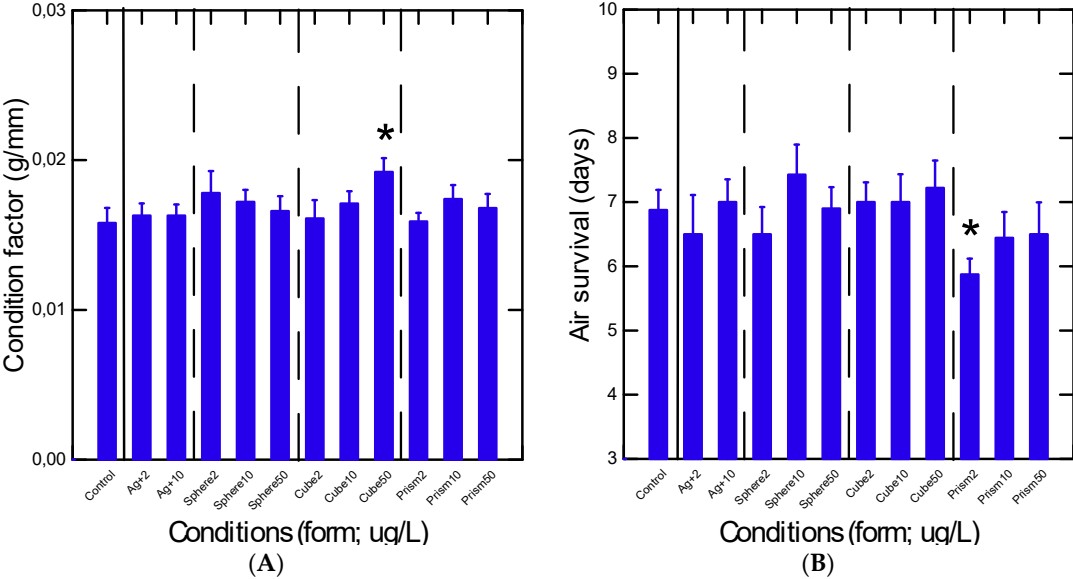

**Figure 1.** Condition factor and air survival time of mussels exposed to nAg geometries. Mussels were collected for condition factor (**A**) and air survival time (**B**). The data represent the mean with the standard error. The star symbol * indicates significant differences from controls.

The biochemical properties of the post-mitochondrial fraction ($15{,}000 \times$ g supernatant) were examined by following changes in viscosity, protein aggregation and neutral lipids levels (Figure 2). Viscosity dropped in mussels exposed to the low concentrations of cubic (2 µg/L) and prismatic (2 and 10 µg/L) nAg relative to control mussels (Figure 2A). However, viscosity returned to control values at higher concentrations for cubic and prismatic nAg. The levels in lipids revealed that prismatic nAg reduced lipids at 10 µg/L and marginally ($p < 0.1$) increased lipids at 50 µg/L (Figure 2B). Correlation analysis showed a trend between lipids and viscosity *(r = 0.45)* (Table 2). This suggests that high neutral lipid levels could increase viscosity in the cytoplasm. Although no significant changes were observed in protein aggregation (Figure 2C), the levels tended to decrease for cubic nAg and were significantly correlated with neutral lipids *(r = 0.65)* and viscosity *(r = 0.23)* (Table 2). No significant differences between nAg of similar SA (10 µg/L spherical and 50 µg/L prismatic nAg) for viscosity, lipids and protein aggregation suggests that no other factors than SA contributed to the observed responses.

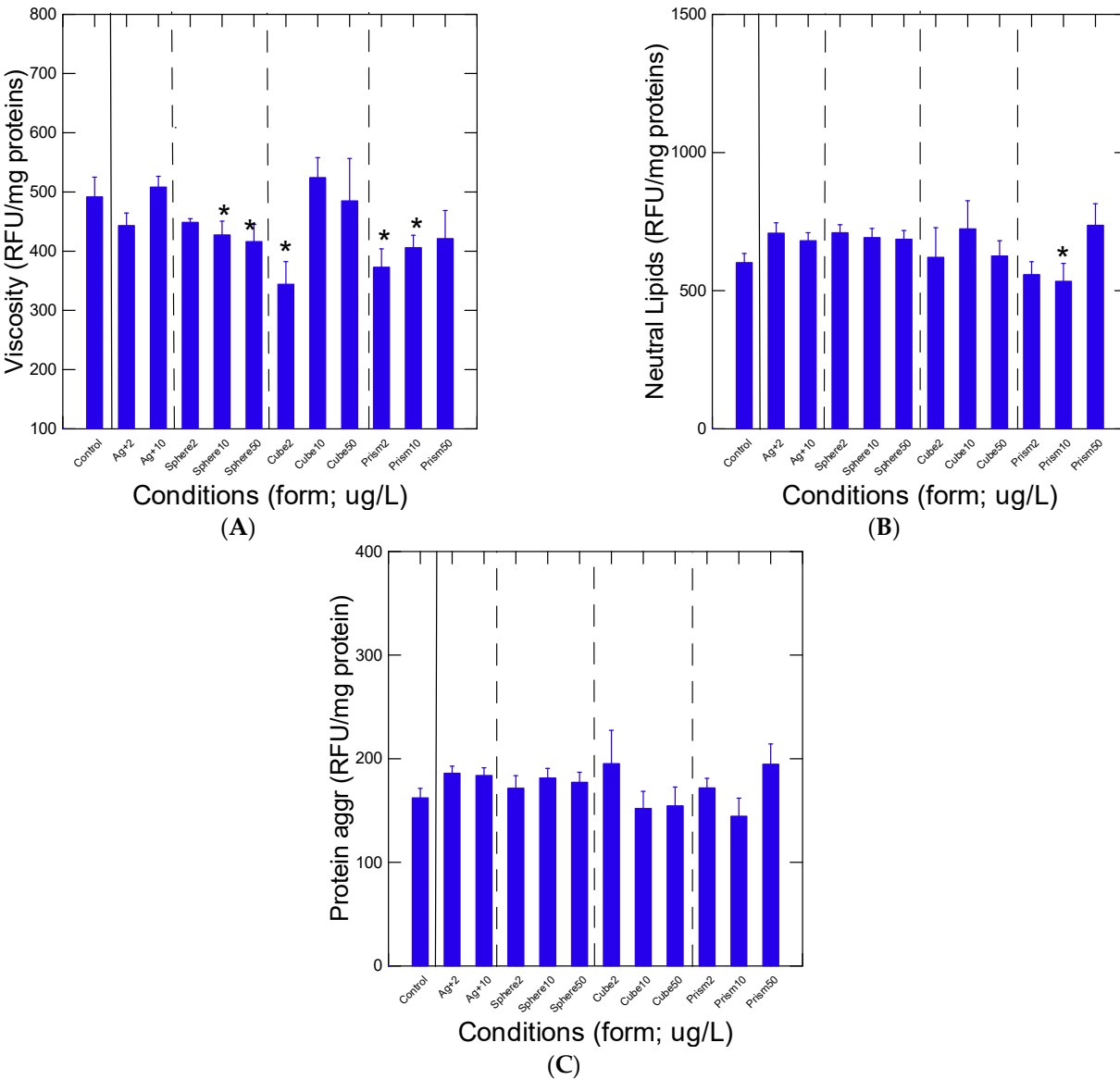

**Figure 2.** Changes in cytoplasm properties in mussels exposed to nAg forms. Mussel whole tissue homogenate fractions were isolated and analyzed for viscosity (**A**), protein aggregation changes (**B**) and neutral lipids (**C**). The data represent the mean with the standard error. The star symbol * indicates significant differences from controls.

**Table 2.** Correlation analysis of biomarker data.

|  | CF | Air Surv. | AChE | FD | Free Zn | aCOX | LPO | Prot. Agg. | Lipids |
|---|---|---|---|---|---|---|---|---|---|
| CF | 1 | | | | | | | | |
| Air surv | 0.08 | 1 | | | | | | | |
| AChE | **0.22** | −0.04 | 1 | | | | | | |
| FD | 0.04 | 0.06 | −0.09 | 1 | | | | | |
| Free Zn | 0.15 | −0.04 | −0.02 | 0.13 | 1 | | | | |
| aCOX | 0.01 | 0.03 | 0.20 | 0.04 | 0.13 | 1 | | | |
| LPO | 0.18 | 0.18 | **−0.22** | 0.00 | 0.01 | −0.10 | 1 | | |
| Prot agg | −0.08 | −0.11 | −0.05 | −0.02 | −0.05 | 0.04 | 0.13 | 1 | |
| Lipids | 0.05 | −0.06 | 0.06 | −0.01 | −0.05 | 0.11 | 0.16 | **0.65** | 1 |
| Viscosity | −0.1 | −0.04 | 0.17 | 0.03 | **0.26** | 0.17 | −0.02 | **0.23** | **0.45** |

Significant correlations are highlighted in bold ($\alpha = 0.05$).

Oxidative stress was determined by measuring labile Zn levels, aCOX activity and LPO (Figure 3A–C). The levels of labile Zn were significantly decreased at 2 µg/L spheric and cubic nAg (Figure 3A) but returned to control levels at higher concentrations, suggesting an hormetic type of effect of nAg, perhaps due to SA-mediated effects. The increased levels of Zn for the 50 µg/L cubic nAg were marginal ($p = 0.1$). Labile Zn levels were significantly correlated with viscosity changes ($r = 0.26$), suggesting that increased viscosity was partly associated with the mobilization of Zn in the cytoplasm (oxidative stress?). The activity in aCOX, a marker of inflammation and oxidative stress, did not change with either the exposure concentration or the forms of Ag (Figure 3B). No significant changes in LPO were observed in mussel tissues (Figure 3C). No significant difference was obtained between nAg of similar SA, with LPO and COX activities suggesting no influence by factors other than SA on the observed responses.

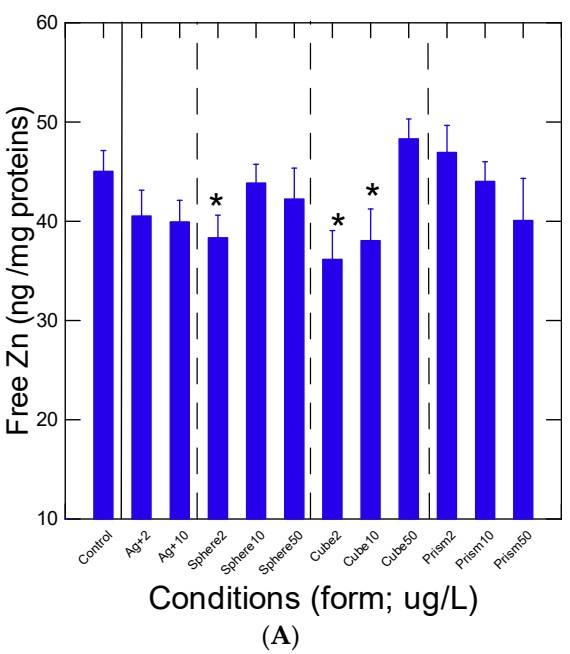

(**A**)

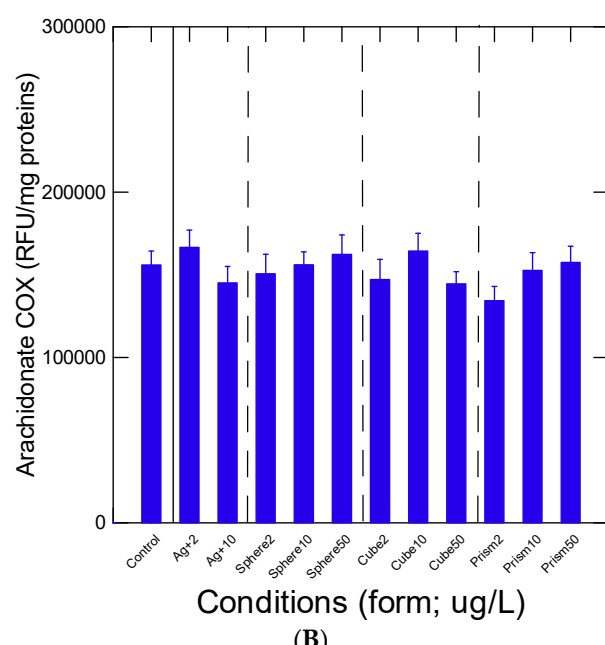

(**B**)

**Figure 3.** *Cont.*

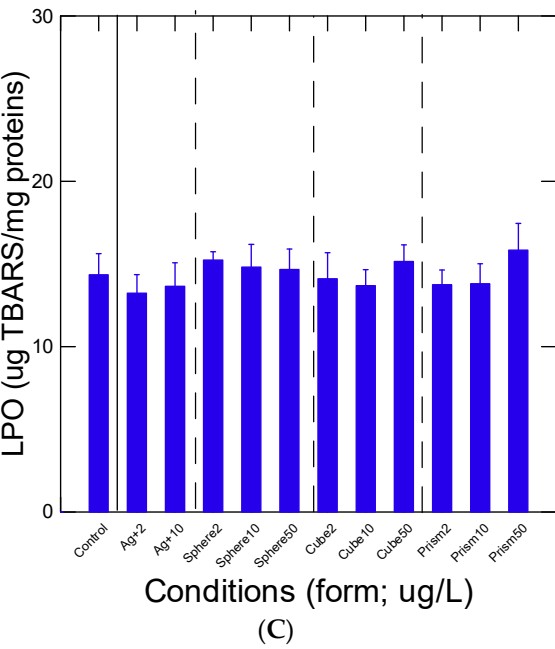

**Figure 3.** Oxidative stress in mussels exposed to various nAg forms. Mussel tissue homogenates were prepared for viscosity (**A**), arachidonate cyclooxygenase activity (**B**) and lipid peroxidation (**C**). The data represent the mean with the standard error. The star symbol * indicates significance from controls.

Neural activity was also examined in mussels treated with the various forms of Ag using AChE activity assessments (Figure 4). A decrease in AChE was observed in most forms of Ag in the following order: prism ($p < 0.001$) > cube ($p = 0.001$) > sphere ($p = 0.01$), and the dissolved Ag was similar to controls ($p = 0.11$) based on statistical differences of the responses (all concentrations included) with respect to controls. Given the possibility that nAg forms (spheres, cubes and prisms) could alter the biophysical properties, such as crowding of the intracellular space, a long-term memory analysis was used to determine the fractal dimension *fD* of the AChE reaction (Figure 4B). The analysis revealed that the *fD* of the AChE reaction was reduced by the various geometries (spheres, cubes and prisms) of nAg compared to controls and to dissolved Ag. With respect to controls, the reduction in the *fD* was, in order of decreases: sphere ($p = 0.01$) < cube ($p = 0.03$) < prism ($p = 0.05$) where the dissolved Ag did not differ from the controls ($p = 0.16$). Based on nAg with similar SA, AChE activity was not affected by spherical and prismatic nAg of similar SA, but a significant effect ($p = 0.05$) was observed for *fD*, suggesting that changes in the *fD* of AChE was influenced by other factor(s) than solely SA (such as form or reactivity).

In the attempt to gain a global understanding of the influence of Ag forms on mussel tissues, a discriminant function and decision tree analysis were performed (Figure 5A,B). The analysis revealed that Ag forms produce characteristic effects based on the biomarkers examined in this study. Discriminant function analysis revealed that most forms of Ag were somewhat discriminated between groups but with a mean classification performance of 35% only. The following endpoints explained most of the variance (93%): AChE, *fD*, viscosity, lipids, labile Zn and viscosity. If we take these and perform a decision tree analysis, the following classification rules are obtained: if viscosity >1.1-fold of controls then it is a cubic nAg; otherwise, if protein aggregation <0.85 of controls then it is a cubic nAg; otherwise, if protein aggregation >1.2-fold of controls then it is a prismatic nAg; otherwise, if the *fD* of AChE < controls then sphere nAg otherwise it is dissolved Ag. Hence, cubic nAg is a compound producing a stronger increase in viscosity than the other forms of Ag in mussels. A prismatic form of nAg leads to more important changes in protein aggregation, while spheric nAg decreases the *fD* of AChE activity with low changes in protein aggregation and

viscosity. The dissolved fraction of Ag produces only slight changes in lipids and protein aggregation.

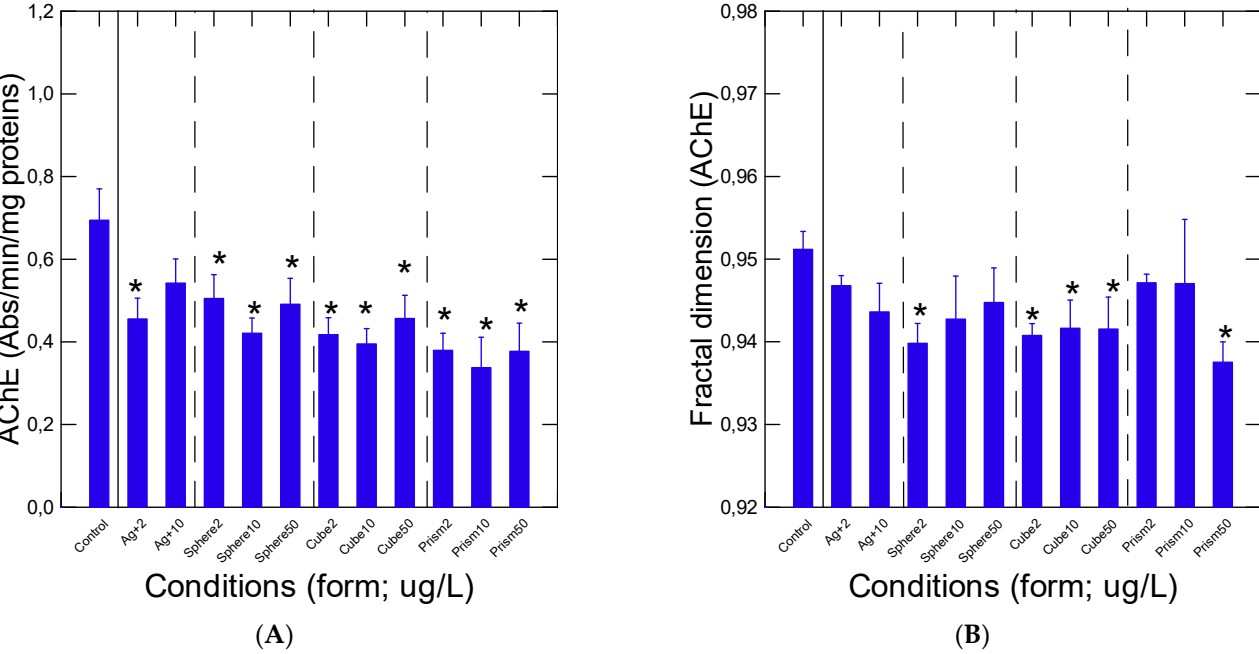

(**A**)  (**B**)

**Figure 4.** Change in AChE activity in mussels exposed to the various Ag forms. Mussel whole tissue homogenate fractions were isolated and analyzed for AChE activity (**A**) and the fractal dimension of AChE (**B**). The data represent the mean with the standard error. The star symbol * indicates significant differences from controls.

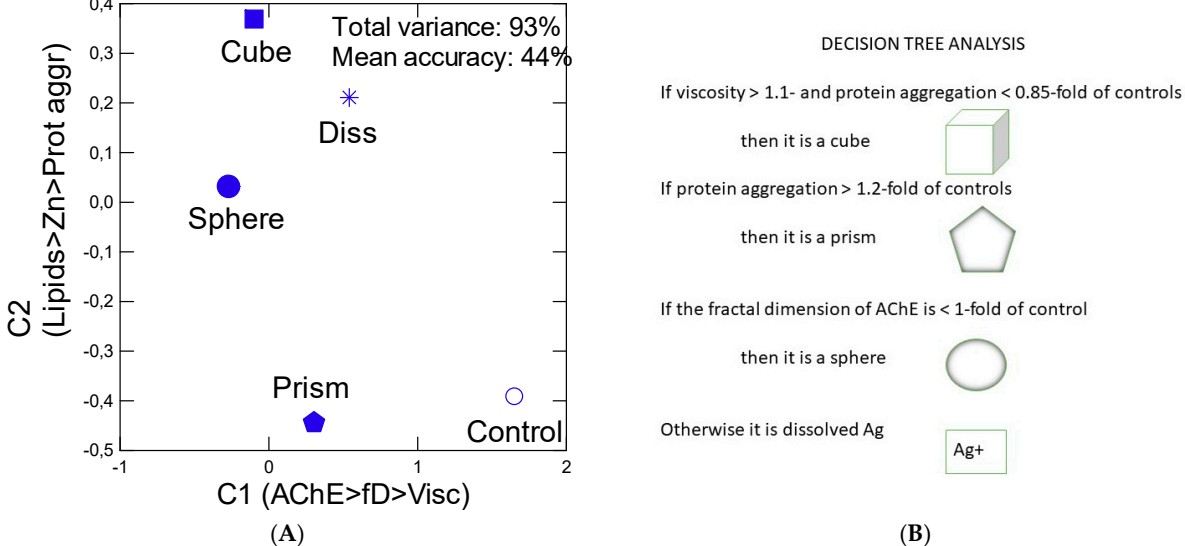

(**A**)  (**B**)

**Figure 5.** Discriminant function and decision tree analysis of biomarker data. The discriminant function analysis (**A**) was performed to distinguish the response patterns between the forms of Ag and the decision tree analysis was performed to determine how the specific effects could discriminate between forms. The total explained variance was 95%, with a mean classification accuracy of 45%. The mean score values are shown for controls, dissolved (Diss) and the three forms of nAg: cubes, spheres and prisms. The decision tree analysis (**B**) goes as follows: start as cubes; if viscosity >1.1-fold of controls then cubic nAg; otherwise, if protein aggregation <0.85 of controls then cube; otherwise, if protein aggregation >1.2-fold of controls then prismatic nAg; otherwise, if *fD* of AChE < controls then sphere otherwise it is dissolved.

## 4. Discussion

The activity of AChE was readily inhibited by all forms of Ag in the present study. Moreover, inhibition appeared stronger with prismatic and cubic nAg compared to spheric and dissolved Ag, thus allowing the rejection of the null hypothesis that all forms of Ag are equally toxic. Studies comparing the toxicity of Ag forms in aquatic organisms are scarce. In a previous study of PVP-coated nAg of different geometries (spheres, cubes and plates), the activity of pyruvate kinase–lactate dehydrogenase was altered, with corresponding changes in the fractal dimension of the enzyme complexes in exposed mussels [10]. In the present study, decreases in the *fD* of AChE suggest that the various forms of nAg change the spatial organisation of the enzyme AChE, perhaps by crowding effects. Other enzymes could be affected as well, not only at the neurological level but in other critical physiological systems. Strong inhibition of AChE activity was also observed in the marine clam *Ruditapes philippinarum* exposed to nAg [14]. Interestingly, while the addition of dissolved organic matter reduced bioavailability and oxidative stress, AChE activity was still affected, suggesting susceptibility of bivalves towards nAg at the neurological level. Disturbance in the cholinergic system in mussels, as determined by AChE activity inhibition, was observed with other nanoparticles, such as gold-coated $nTiO_2$ [27]. The decrease was accompanied with increased oxidative stress, as determined by catalase, glutathione S-transferase activities and LPO levels in tissues. Behavior was seemingly affected by reduced filtration activity (clearance rate of neutral red particles).

In the present study, oxidative stress was not observed given that the nanoparticles were PVP-coated and the surfaces were equally reactive [28]. Although PVP-coated nAg was more bioavailable than citrate-coated nAg, no evidence of oxidative stress was found in the digestive gland and gill tissues. Neurological impacts were also found in the snail *Lymnea stagnalis* exposed to uncoated 100 nm diameter nAg [29]. A biphasic response in memory function was observed where enhanced memory formation was observed at 10 µg/L followed by reduced memory at 50 µg/L. In another study, *Scorbicularia plana* clams exposed to gold nanoparticles (nAu) also displayed behavioral changes [30]. This study showed increased oxidative stress, including metallothionein induction in clams exposed to relatively high concentrations (100 µg/L nAu) of different sizes leading to decreased burrowing activity. Interestingly, not all sizes of nAu displayed similar effects, suggesting that size could influence toxicity. For example, metallothioneins were induced with sizes of 5 and 40 nm, while catalase activity was increased by 15 and 40 nm diameter nAu.

This suggests that nanoparticles induce size-dependent interaction in tissues. The influence of the size of nAg revealed the importance of protein adsorption at surfaces or corona formation [31]. The curvature of the nanoparticle is stronger in smaller nanoparticles, favoring more stable binding of protein coronae. Stronger binding arises from disruption to the secondary structure of the proteins via cysteine–cystine interactions with Ag at the surface. The stronger curvature found in smaller sizes of nAg (high surface volumes) contributes to a more stable protein corona, thus limiting availability towards cells and toxicity from the release of ionic Ag in cells compared to uncoated nAg. If we extend this reasoning with other geometries, those producing more curvatures (higher surface areas) should produce more stable coronae and be less toxic in mussels. Indeed, inhibition of AChE was less severe with spheric and cubic nAg (high SA, Table 1) than prismatic nAg (lowest SA and curvature). With respect to viscosity, spheric nAg decreased viscosity in a concentration-dependent manner, but viscosity returned to control values at higher concentrations for cubic and prismatic nAg. The reasons for these non-linear changes are unclear but it could be related to SA-induced crowding effects. In marine clams, AChE activity was also decreased in the gills and digestive glands of clams exposed to $nTiO_2$ and Au-coated $nTiO_2$ [32]. This decrease was associated with decreased feeding activity and filtration rates. While catalase was induced by either $nTiO_2$ or Au-coated $nTiO_2$, oxidative damage (LPO) was more strongly expressed with uncoated $nTiO_2$. Dissolved but not nAu-inhibited AChE in the hemolymph of *Mytilus galloprovinciallis* suggests that ionic forms of the metal acted on the enzyme [33]. Inhibition of AChE in fish exposed to nAg

was observed in muscles and brains [34]. However, the contribution of dissolved Ag was not examined in this study.

Nanoparticles are recognized to produce effects not only from the release of dissolved elements but from other properties, such as form, size and coatings [35]. The capacity of nanoparticles to absorb existing chemicals, such as PAHs and other chemicals, e.g., pharmaceuticals or pesticides, is another property of nanomaterials referred to collectively as "Trojan horse effects" [36,37]. Surface properties could also be changed by surface interactions, as shown above, involving the adsorbtion of proteins, lipids or other macromolecules (natural organic matter) at the surface of nanoparticles [33]. The formation of protein or lipid coronae at the surface of nAg was shown to preserve the Ag core, thereby limiting rapid diffusion of ionic $Ag^+$ into the medium and toxicity [38]. Serum bovine albumin binds at the surface of nAg, forming a corona which, in turn, could interact with lipid vesicles [39]. The fluidity of lipid vesicles was increased in the presence of albumin-coated nAg compared to uncoated nAg. This is in agreement with the observation of decreased viscosity in found nAg but not with ionic/dissolved Ag in the present study. Moreover, decreased viscosity was stronger with lower SA prismatic nAg compared to spheric and cubic nAg (Table 1). This suggests that weakened protein coronae increase interactions with lipid vesicles, perhaps by hydrophibic interactions of the PVP-coating of nAg. Based on correlation analysis, viscosity was significantly correlated with lipids, protein aggregation and labile Zn (Table 2). It is noteworthy that labile Zn levels and viscosity changes followed a non-linear U-shaped curve, suggesting that the observed changes occur in a biphasic manner. For example, the drop in viscosity at low concentrations of cubic and prismatic nAg could be associated with changes in protein–lipid interactions at the surface, where viscosity is returned to control values as the concentration of nAg increases in the cytoplasm. These biophysical changes and labile Zn changes were shown to induce protein denaturation and turnover in cells exposed to nAg [40], which could influence viscosity. This suggests that the geometry and corona surfaces could influence the availability, the biophysical characteristics of the cytoplasm and the toxicity of nAg. Indeed, various forms of nAg in the same size range were shown to induce partially organized nematic liquid crystals in the cytoplasm, leading to increased protein denaturation and turnover in cells [10].

Another consequence with respect to these interactions caused by the fine nanoparticles is the reduction of free space in the cytoplasm, in which complex macromolecules interact during metabolism. As shown in this study, the *fD* of AChE was reduced, thereby slowing down reaction rates at high substrate concentrations, perhaps by crowding (traffic) by nanoparticles or changes in liquid crystals, as evidenced by viscosity and lipid mobilisation. Decreased enzyme activity by steric interactions was also shown with the pyruvate kinase–lactate dehydrogenase complex, where various forms of nAg in mussels reduced the (fractal) dimension of the enzyme complex activities [10]. This phenomenon (i.e., molecular crowding) was also observed with polystyrene plastic nanoparticles with lactate dehydrogenase activity in mussels and hydra [40]. Silver nanoparticles could also interact more directly with enzymes following removal of the coronae or surface coatings [41]. The enzyme involved in the catabolism of catechols (dopamine, adrenalin) catechol O-methyltransferase forms a corona on nAg, leading to tryptophan fluorescence quenching. It was shown that the interaction of the enzyme at the surface of nAg was electrostatic in nature (the surface of nAg would contain positive charges from $Ag^+$), leading to the structural change of the enzyme and reducing its activity.

In conclusion, exposure of mussels to four forms of Ag (dissolved, cubic, spheric and prismatic) led to decreased viscosity, labile Zn and AChE activity in mussel tissues. The decrease in viscosity was associated with lipid and protein aggregation in the homogenate fraction of exposed mussels. The intensity in AChE activity inhibition depended on the form where primastic > cubic > spheric nAg and on dissolved Ag. Both spheric and cubic nAg more strongly decreased the *fD* of AChE where cubic nAg increased viscosity and decreased protein aggregation in the cytoplasm. Prismatic nAg also increased both

protein aggregation and viscosity with fewer changes in AChE. The toxicity of nAg to freshwater mussels was therefore dependent on geometry in addition to surface properties and concentrations.

**Author Contributions:** Conceptualization, J.A. and F.G.; methodology, J.A. and C.P.; software, J.A. and C.P.; validation, K.J.W. and F.G.; formal analysis, F.G.; investigation, F.G. and J.A.; resources, K.J.W. and F.G.; data curation, J.A., and C.P.; writing—original draft preparation, F.G.; writing—review and editing, F.G. and K.J.W.; visualization, all authors; supervision, F.G. and K.J.W.; project administration, F.G.; funding acquisition, F.G. All authors have read and agreed to the published version of the manuscript.

**Funding:** This research received no external funding.

**Institutional Review Board Statement:** Not applicable.

**Informed Consent Statement:** Not applicable.

**Data Availability Statement:** Data are available upon demand to the corresponding author.

**Acknowledgments:** The authors acknowledge the financial support of the Chemical Management Plan of Environment and Climate Change Canada.

**Conflicts of Interest:** The authors declare no conflict of interest.

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
