# Peer review of "The Influence of Silver Nanoparticle Form on the Toxicity in Freshwater Mussels"

_applsci, doi:10.3390/app12031429_

Round 1

Reviewer 1 Report

Review of “The Influence of Silver Nanoparticle Form on the Toxicity in Freshwater Mussels” for Applied Sciences, January 2022

Summary:

This paper looked at the effect of different shapes of silver nanoparticles (sphere, cube and prism) on various biomarkers and acute toxicity to Dreissena bugensis mussels. Overall, the paper contains a lot of data, but the data sets aren’t discussed well and the paper itself is poorly organized and confusing. All of that can be fixed, but to me the biggest issue is that the “biomarker” endpoints respond in non-dose-dependent fashions, specifically in that some increase in response to the nAg and some decrease, and the authors seem to call every change as “toxicity”. This conclusion does not make sense to me – how could an increase and decrease in the same endpoint both result in toxicity? Further, all of the “biomarkers” need to be explained better in terms of what physiological connection they have to the whole health of the organism – what does it mean for the viscosity to change and why would we care about these changes? Lastly, this same group of authors published a very similar paper end of last year (reference 10 in this manuscript) which seems to publish a very similar data set and some of the text from this manuscript is directly plagiarized from that published paper. This is a grave concern and needs to be addressed.

 Abstract:

- There are some minor typos that need to be addressed (e.g. closing parentheses that has no opening in L17 and this phrase is missing a verb: “spheric nAg product more important changes towards the fractal dimension of AChE” (L17-18)”

Introduction:

- Overall there are some small issues with the English writing in this manuscript, e.g. subject verb agreement (L26 “the nanotechnology industry have grown” – should be “has”) and minor typos – nothing major and this didn’t influence my review of the manuscript overall, but needs to be addressed before publication.

- L27: this reference is from 2010 but the authors say that the nanotechnology industrial has been growing “over the last decade” – can the authors find a more recent reference to support this statement?

- L26 – 81 this is one paragraph and over half of the introduction! It’s way too long and covers too many topics. Rewrite this section such that each paragraph covers each major topic and there are connections between them.

- L60: this reference (10) is by the same authors and seems to be the same or a very similar dataset. The authors need to very carefully make clear how this publication is different from the work already published. I think the authors are reporting different endpoints, but the papers need to be clearly distinguished. For instance, Tables 1 and 2 seem to be almost identical between the two studies, differing in what data are reported specific for the study, but are set up the same way. It appears to me the authors are trying to get multiple papers out of a single dataset, which is fine as long as each dataset can stand on its own as its own publication and that the distinctions between papers is made very clear.

 - L89: a more general discussion of the “protein corona” needs to be included – what is this exactly and why is it important? Is this a protein all organisms have? What are the implications of “changes in cells such as protein corona”?

Methods:

- L110-119 are identical to the published paper (Auclair et al. 2021) – this is plagiarism and must be revised. It’s likely this is true in other sections and must be carefully vetted – this is just where I noticed it.

- Table 1 in this submitted manuscript is almost identical to the published, very similar study (Auclair et al. 2021); one large difference is that the sizes of the all of the particles are different but the surface areas are the same – how is that possible?

Results:

- L227-229: why was the nominal concentration “determined after 1 h dissolution” when the exposures were 96 hours? The authors need to include whatever data and rationale they have for this to be a safe assumption.

- L255-268 and Figure 2: so the only big effect was on viscosity and it only decreased in a dose-dependent manner for the spheres. For all other shapes, viscosity increased as the concentration increased – why? That to me indicates that these endpoints aren’t responding in the way the authors expect and maybe aren’t a good endpoint to assess toxicity or physiological effect of a contaminant.

- L261-262: What does it mean that viscosity and lipid levels are correlated? Is this expected?

- Table 2: the table says that “significant correlations are highlighted in bold” but none are highlighted. The paper later goes on to discuss “significant” correlations so presumably some are significant?

- L272-282: Same comment as above for labile Zn, except for this endpoint, none of the responses follow a traditional dose-depedence. The treatments with the largest effects were for the lowest concentration within each shape treatment. This is also slightly concerning to me that this endpoint or maybe method (I don’t have any experience with these methods) are not appropriate.

- L276-7: again, why is it significant that these are correlated? Why were the authors looking for correlations between these biomarkers?

- L313-314: should this be a comma instead of a period? Do these two sentence connect?

Discussion:

- L332-367: this is a very long paragraph. Again, separate into multiple paragraphs by main points.

- L339: again I think it needs to be more clear that this paper is another study by the same authors. So instead of saying “in a previous study”, say something like “our group previously found” or something similar to make this clear.

- L344-347: this paragraph is disorganized and unclear, so I can’t tell, but is this sentence about ref 14 or the current study?

- L370-378: none of these sentences have citations, is this information all from ref 30, cited at the beginning of this paragraph? More citations are needed. I understand some of this are probably new hypotheses the authors are coming up with based on their data, but it needs to be clear what part(s) are established in the literature and what part(s) are new.

- L380-381: following on my comments on the section about viscosity in the results section, viscosity increased and decreased in response to the different forms and concentrations of nAg. Specifically, the authors here are discussing prismatic nAg, for which the viscosity increased with increasing concentration. What does that mean? The authors here discuss “changes” in viscosity, seemingly acknowledging that not all the changes were negative, but they never directly address increases versus decreases in viscosity and what that means biologically.

- L381-390: why is this discussion of past studies of AChE in this paragraph? I realize some of this is size-dependent interactions, but a lot of the discussion isn’t and should be in the previous paragraph about AChE effects found in other studies.

- L391-434: this is another very long paragraph; please revise.

- L404-407: Again, the authors need to explain how the viscosity could be decreased and increased by nAg before using this data to draw conclusions.

- L411: none of the correlations in Table 2 are indicated as significant

- L435-437: again, exposure led to decreased and increased viscosity and labile Zn, so stating this here as a “conclusion” is too strong of a statement unless these data are better explained. The only clear effect was on AChE activity.

- L441-442 the authors here finally discuss the increases in some of these endpoints, but as part of the conclusion rather than in the discussion or otherwise try to explain what these endpoints mean.

Author Response

Please also see attachment.

--

Answers to comments are provided in blue here and in the manuscript for easy tracking.

Summary

This paper looked at the effect of different shapes of silver nanoparticles (sphere, cube and prism) on various biomarkers and acute toxicity to Dreissena bugensis mussels. Overall, the paper contains a lot of data, but the data sets aren’t discussed well and the paper itself is poorly organized and confusing. All of that can be fixed, but to me the biggest issue is that the “biomarker” endpoints respond in non-dose-dependent fashions, specifically in that some increase in response to the nAg and some decrease, and the authors seem to call every change as “toxicity”. It is not usual that biomarkers respond in U shaped dose response curve, this point will be raised in the discussion for better clarity. We checked the dose response curves and there was no non-linear responses at the significance level (p<0.05). For example, ACHE changes were decreased significantly by some forms of nAg with no significant increases in respect to controls.

This conclusion does not make sense to me – how could an increase and decrease in the same endpoint both result in toxicity? Further, all of the “biomarkers” need to be explained better in terms of what physiological connection they have to the whole health of the organism – what does it mean for the viscosity to change and why would we care about these changes? This was included in the introduction at lines 92-95.

Lastly, this same group of authors published a very similar paper end of last year (reference 10 in this manuscript) which seems to publish a very similar data set and some of the text from this manuscript is directly plagiarized from that published paper. This is a grave concern and needs to be addressed. I will double check on this, we did examine the influence on the geometry with other end points but this was another set of experiments/exposure with other endpoints. There is no plagiarism this is a following study on the issue of toxicity of various forms of nanoparticles. This was clearly explained in the introduction lines 60-68 but I added more information on the toxicity at lines 68-69, 70-71 and 77-78.

 Abstract:

There are some minor typos that need to be addressed (e.g. closing parentheses that has no opening in L17 and this phrase is missing a verb: “spheric nAg product more important changes towards the fractal dimension of AChE” (L17-18)” This was corrected

Introduction:

-Overall there are some small issues with the English writing in this manuscript, e.g. subject verb agreement (L26 “the nanotechnology industry have grown” – should be “has”) and minor typos – nothing major and this didn’t influence my review of the manuscript overall, but needs to be addressed before publication. OK the grammar was revised

L27: this reference is from 2010 but the authors say that the nanotechnology industrial has been growing “over the last decade” – can the authors find a more recent reference to support this statement? Done

L26 – 81 this is one paragraph and over half of the introduction! It’s way too long and covers too many topics. Rewrite this section such that each paragraph covers each major topic and there are connections between them. Done.

L60: this reference (10) is by the same authors and seems to be the same or a very similar dataset. The authors need to very carefully make clear how this publication is different from the work already published. I think the authors are reporting different endpoints, but the papers need to be clearly distinguished. For instance, Tables 1 and 2 seem to be almost identical between the two studies, differing in what data are reported specific for the study, but are set up the same way. It appears to me the authors are trying to get multiple papers out of a single dataset, which is fine as long as each dataset can stand on its own as its own publication and that the distinctions between papers is made very clear. As discussed above, this study followed the previous one in order to examine more closely the effects of the form of nanoAg in mussels. We therefore repeated the exposure and reanalyzed mussels for other endpoints presented in this manuscript. There are no duplication, we repeated the experiments and looked at other endpoints such as neurotoxicity by acetylcholinesterase activity.

L89: a more general discussion of the “protein corona” needs to be included – what is this exactly and why is it important? Is this a protein all organisms have? What are the implications of “changes in cells such as protein corona”? This term was removed in the introduction and was the formation of protein coatings on nanoparticles was discussed in the discussion.

Methods:

- L110-119 are identical to the published paper (Auclair et al. 2021) – this is plagiarism and must be revised. It’s likely this is true in other sections and must be carefully vetted – this is just where I noticed it. The sentences were rewritten to remove this impression.

Table 1 in this submitted manuscript is almost identical to the published, very similar study (Auclair et al. 2021); one large difference is that the sizes of the all of the particles are different but the surface areas are the same – how is that possible? Yes it is logical, the chemical behavior do not overly changes in the exposure media, it should not change between different exposure experiments in the same laboratory conditions and water. The minute changes are probably from a different batch or lot of nAg.

Results:

- L227-229: why was the nominal concentration “determined after 1 h dissolution” when the exposures were 96 hours? The authors need to include whatever data and rationale they have for this to be a safe assumption. The measurement was made after one h to show that nAg were stable in suspension and thus confirmed exposure to mussels. This was added at lines 232-33.

- L255-268 and Figure 2: so the only big effect was on viscosity and it only decreased in a dose-dependent manner for the spheres. For all other shapes, viscosity increased as the concentration increased – why? That to me indicates that these endpoints aren’t responding in the way the authors expect and maybe aren’t a good endpoint to assess toxicity or physiological effect of a contaminant. Yes, viscosity was used as another endpoint for the influence of the form of nAg on cytoplasm viscosity. The reasons for why the changes were reduced at higher concentration for prismatic and cubic are unclear for the present but this data was still worth showing since the more linear effects of spheric nAg. See lines 385-88 in the discussion.

- L261-262: What does it mean that viscosity and lipid levels are correlated? Is this expected? Decreased viscosity could occur by changes in lipids. See lines 267-68.

Table 2: the table says that “significant correlations are highlighted in bold” but none are highlighted. The paper later goes on to discuss “significant” correlations so presumably some are significant? This was added in the revised manuscript.

- L272-282: Same comment as above for labile Zn, except for this endpoint, none of the responses follow a traditional dose-depedence. The treatments with the largest effects were for the lowest concentration within each shape treatment. This is also slightly concerning to me that this endpoint or maybe method (I don’t have any experience with these methods) are not appropriate. U shaped dose response are usual in biomarkers, in some cases effects are higher at lower concentrations and saturation/exhaustion of the response are found at higher concentrations.

- L276-7: again, why is it significant that these are correlated? Why were the authors looking for correlations between these biomarkers? This was added at lines 285-86.

L313-314: should this be a comma instead of a period? Do these two sentence connect?

Discussion:

- L332-367: this is a very long paragraph. Again, separate into multiple paragraphs by main points.Done

- L339: again I think it needs to be more clear that this paper is another study by the same authors. So instead of saying “in a previous study”, say something like “our group previously found” or something similar to make this clear. Ok done.

- L344-347: this paragraph is disorganized and unclear, so I can’t tell, but is this sentence about ref 14 or the current study? We added some clarification at lines 347-49.

- L370-378: none of these sentences have citations, is this information all from ref 30, cited at the beginning of this paragraph? More citations are needed. I understand some of this are probably new hypotheses the authors are coming up with based on their data, but it needs to be clear what part(s) are established in the literature and what part(s) are new. All this part was from ref 40, we clarified this at line 370.

- L380-381: following on my comments on the section about viscosity in the results section, viscosity increased and decreased in response to the different forms and concentrations of nAg. Only the decrease was significant hence no biphasic responses as explained above.

Specifically, the authors here are discussing prismatic nAg, for which the viscosity increased with increasing concentration. What does that mean? The authors here discuss “changes” in viscosity, seemingly acknowledging that not all the changes were negative, but they never directly address increases versus decreases in viscosity and what that means biologically. This was corrected, viscosity only significantly dropped but returned to control values as explained above.

- L381-390: why is this discussion of past studies of AChE in this paragraph? I realize some of this is size-dependent interactions, but a lot of the discussion isn’t and should be in the previous paragraph about AChE effects found in other studies. Do not understand, this section discuss changes on AChE by other and forms of nanoparticles. It is not misplaced?

- L391-434: this is another very long paragraph; please revise. It was reduced.

- L404-407: Again, the authors need to explain how the viscosity could be decreased and increased by nAg before using this data to draw conclusions. See above…..

- L411: none of the correlations in Table 2 are indicated as significant. This was added in the revision.

- L435-437: again, exposure led to decreased and increased viscosity and labile Zn, so stating this here as a “conclusion” is too strong of a statement unless these data are better explained. The only clear effect was on AChE activity. This was all changed in the revision. I think the confusion was decreased by increased changes in viscosity and Zn but was rather a significant decrease and returned to control values afterwards….

- L441-442 the authors here finally discuss the increases in some of these endpoints, but as part of the conclusion rather than in the discussion or otherwise try to explain what these endpoints mean. Special effort was provided to clarify this better as shown above.

Thank you for the comments.-author reply

Reviewer 2 Report

The manuscript "The Influence of Silver Nanoparticle Form on the Toxicity in Freshwater Mussels" tries to examine the toxicity of various shapes of nAg (spherical, cubic and prismatic) and ionic Ag+ in the tissues of freshwater mussels. To determine the toxic effect of nAg the authors was examined AChE activities, oxidative stress and protein aggregation. It was a good article and the results add some new data of the effect of AgNPs on the freshwater mussel to the present knowledge. However, there are some r part of manuscript  to be improved as well.

  • An improvement of experimental plan is needed. In the text it’s not clear how were prepared the concentrations of 2 μg/L of Ag nanoparticles and the concentrations of 2 and 10 μg/L of ionic Ag. I suggest to clarify.
  • The use of nanoparticles from different type means the authors introduce the problem that differences in nanoparticle composition may be affecting the observed changes. Though the authors describe characterization of the particles to alleviate this concern, this data needs to be more clearly described in the paper.
  • For determination of AChE activities, the authors use the soft tissue of mussels but in the results and discussion they say that was examined mussel digestive gland. In general the most sensitive response of AChE in mussels soft tissue are gills. I suggest to clarify in the text the exact tissue utilized and why.
  • Data analysis report only two concentrations. The authors talk about three concentrations for nAg particles types. Please improve data analysis.

Round 2

Reviewer 1 Report

Thank you for your editing. I'm still concerned about possible plagiarism between this manuscript and the currently published manuscript - the fact that the sentences in which I identified plagiarism was corrected is great but I worry that other sections, especially in the methods, may still be plagiarized from the currently published version. The authors assert that "There is no plagiarism this is a following study on the issue of toxicity of various forms of nanoparticles" but you can self-plagiarize by using the same words as a paper that is published, even if written by the same authors. Even if the papers are about completely different studies, self-plagiarism is still an issue if you re-use text or very similar text. I urge the authors to do a more thorough review of this text to ensure there are not more places in which they have re-used exact or even very similar text.

My only other request is that the authors put in the discussion of the U-shaped response of biomarkers somewhere in the text - they made that argument multiple times in the response and it's a compelling argument that needs to be in the text with supporting references.

Author Response

Thank you for your editing. I'm still concerned about possible plagiarism between this manuscript and the currently published manuscript - the fact that the sentences in which I identified plagiarism was corrected is great but I worry that other sections, especially in the methods, may still be plagiarized from the currently published version. The authors assert that "There is no plagiarism this is a following study on the issue of toxicity of various forms of nanoparticles" but you can self-plagiarize by using the same words as a paper that is published, even if written by the same authors. Even if the papers are about completely different studies, self-plagiarism is still an issue if you re-use text or very similar text. I urge the authors to do a more thorough review of this text to ensure there are not more places in which they have re-used exact or even very similar text. The text was reformulated in sectors of high correspondence of the wording. It should be corrected now.

My only other request is that the authors put in the discussion of the U-shaped response of biomarkers somewhere in the text - they made that argument multiple times in the response and it's a compelling argument that needs to be in the text with supporting references. Done see lines 407-414.